# Investigating the Effect of Tyrosine Kinase Inhibitors on the Interaction between Human Serum Albumin by Atomic Force Microscopy

**DOI:** 10.3390/biom12060819

**Published:** 2022-06-11

**Authors:** Yuna Fu, Jianhua Wang, Yan Wang, Heng Sun

**Affiliations:** 1Key Laboratory of Biorheological Science and Technology, Ministry of Education, College of Bioengineering, Chongqing University, Chongqing 400044, China; 201719021086@cqu.edu.cn (Y.F.); sunheng@cqu.edu.cn (H.S.); 2Key Laboratory of Drug Design, College of Chemistry and Chemical Engineering, Huangshan University, Huangshan 245041, China; wangyan1992@cqu.edu.cn

**Keywords:** protein–protein interactions, human serum albumin, atomic force microscopy, tyrosine kinase inhibitors

## Abstract

It is important for elucidating the regulation mechanism of life activities, as well as for the prevention, diagnosis, and drug design of diseases, to study protein–protein interactions (PPIs). Here, we investigated the interactions of human serum albumin (HSA) in the presence of tyrosine kinase inhibitors (TKIs: imatinib, nilotinib, dasatinib, bosutinib, and ponatinib) using atomic force microscopy (AFM). The distribution of rupture events including the specific interaction force *F_i_* and the non-specific interaction force *F*_0_ between HSA pairs was analyzed. Based on the force measurements, *F_i_* and *F*_0_ between HSA pairs in the control experiment were calculated to be 47 ± 1.5 and 116.1 ± 1.3 pN. However, *F_i_* was significantly decreased in TKIs, while *F*_0_ was slightly decreased. By measuring the rupture forces at various loading rates and according to the Bell equation, the kinetic parameters of the complexes were investigated in greater detail. Molecular docking was used as a complementary means by which to explore the force of this effect. The whole measurements indicated that TKIs influenced PPIs in a variety of ways, among which hydrogen bonding and hydrophobic interactions were the most important. In conclusion, these outcomes give us a better insight into the mechanisms of PPIs when there are exogenous compounds present as well as in different liquid environments.

## 1. Introduction

It is well-known that protein–protein interactions (PPIs) play a major role in the normal operation of the body, as well as the regulation of various physiological functions [1]. Therefore, the study of PPIs has great theoretical significance for elucidating the molecular basis for diseases, finding potential therapeutic targets, and understanding the molecular mechanisms of biological processes [2]. The interactions between proteins are not only affected by pH and temperature, but also by exogenous compounds [3]. Researchers have used spectroscopy to study the interaction of the tyrosine kinase inhibitors erdafitinib and vandetanib with human serum albumin, and proposed that the introduction of the drug affects the hydrophobic domain of the protein, especially the microenvironment around tryptophan [4,5]. In addition, Wani et al., in their study on the interaction of the tyrosine kinase inhibitor erlotinib with BSA [6], proposed that the presence of the drug affects the formation of protein complexes. These studies suggest that the introduction of drugs alters the active structure of proteins, thereby affecting interactions between proteins, and the alteration of these interactions can further lead to the occurrence of disorders and even diseases in the body [7]. In addition, PPIs are widely present in biological organisms; on the one hand, they maintain the structural stability of proteins and/or other macromolecules, and on the other hand, they are the basis for performing the functions of proteins and other macromolecules [8,9]. As a result, studying the impact of small-molecule drugs on PPIs is extremely important in practice.

Human serum albumin (HSA), one of the most abundant proteins in the blood, is critical to living systems and plays a role in most life processes [10]. HSA has also been extensively studied in previous experiments due to its unique structure and high stability. Imatinib was the first tyrosine kinase inhibitor (TKI) to be approved as a front-line treatment for chronic myeloid leukemia (CML), and its introduction ushered in an era of targeted therapies using TKIs. There have been subsequent developments in TKIs, such as nilotinib, dasatinib, bosutinib, and ponatinib, that reduce the resistance of imatinib and improve the efficiency of treatment [11]. Like other drugs, the distribution and transport of TKIs in the body are mainly dependent on plasma albumin [12]. Therefore, studying the effects of TKIs on protein–protein interactions makes it possible to develop novel anti-leukemia drugs through these interactions.

There are many methods to study PPIs, such as the most commonly used immunoprecipitation (IP) [13], tandem affinity purification-mass spectrometry (TAP-MS) [14], and the pull-down experiment that is increasingly favored by researchers [15]. In addition, fluorescence resonance energy transfer (FRET), surface plasmon resonance (SPR), and optical tweezers are widely used in this research field [16,17,18]. For nanoscale protein assay experiments, AFM is a very powerful tool because it uses AFM tips modified with biomolecules to measure interactions with receptors. In addition, single-molecule dynamic force spectroscopy (DFS) of AFM can provide information on the dynamics of interacting molecules by varying the loading rate [19]. DFS has been successfully employed to address the intermolecular forces that govern biological systems as carbohydrates: proteins [20,21,22], lipid menbranes, flavoenzymes [23], DNA strands [24], RNA strands [25], and lignocellulosic biopolymers [26], among others. In addition, because of the high temporal and spatial resolution of AFM, as well as the enormous force sensitivity required under physiological environmental conditions [27], an understanding of the molecular mechanism of PPIs at a fundamental level is possible when viewed from either a mechanical or dynamic perspective. Therefore, AFM is a suitable candidate method to measure the interactions between HSA pairs at the molecular level.

An AFM technique was employed in this study to examine drug molecules’ effects on PPIs at the nanoscale, to determine the aggregation and mechanical behaviors of HSA in the presence of various tyrosine kinase inhibitors. At the same time, molecular docking was used to summarize the interaction between drugs and proteins, and to help verify the experimental results. The findings reveal that small molecules influence PPIs through a variety of forces, with hydrogen bonding and hydrophobic interactions serving as the primary active force. At the same time, the dissociation constant (*k_off_*) of HSA pairs increased in the presence of TKIs, indicating that adhesion forces between proteins decreased and complexes became less stable. Additionally, molecular docking results prove the interactions between TKIs and HSA. These results may have a certain guiding significance for the development of new drugs and the treatment of diseases.

## 2. Materials and Methods

### 2.1. Materials

Human serum albumin (HSA), 16-Mercaptohexadecanoic acid (MHA), 1-ethyl-3-(dimethylaminopropyl) carbodiimide hydrochloride (EDC), and *N*-hydroxysulfosuccinimide (NHS) were all obtained from Sigma Aldrich. TKIs (imatinib, nilotinib, dasatinib, bosutinib, and ponatinib) were obtained from MedChemExpress (MCE). Gold-coated silicon wafers were obtained from TED PELLA, INC, and AFM probes were supplied by Budget Sensors (Sofia, Bulgaria). Phosphate-buffered saline (10 mM PBS, 140 mM NaCl, 3 mM KCI, pH 7.4) and ethanol (guaranteed grade) were supplied by Merck Co. (Readington Township, NJ, USA), and Millipore water and analytical grade reagents were used throughout the study.

### 2.2. Functionalization of Substrate

Before immobilizing the protein on the substrate, the gold-coated substrate was modified to generate a thiol-containing self-assembled monolayer (SAM) [28]. The 1.0 cm × 1.0 cm gold-coated substrates were immersed in piranha solution (H_2_SO_4_:H_2_O_2_ = 3:1, *v*/*v*) for 30 min to eliminate stains on the surface, then rinsed five times with ultrapure water and anhydrous ethanol alternatively before being blown dry with nitrogen gas. The cleaned substrates were reacted with 1 mM MHA solution for 24 h to form a tightly ordered monomolecular layer on the substrate surface. To remove unbound thiol molecules, substrates were sonicated in anhydrous ethanol for an additional 3 min after incubation. Finally, the modified substrates were alternately rinsed with anhydrous ethanol and ultrapure water three times and dried with nitrogen to prepare for the next step of protein immobilization.

### 2.3. Protein Immobilization

The SAM prepared in the previous step was incubated in a solution of 2 mg/mL NHS and 2 mg/mL EDC for 1 h. During the incubation, the carboxyl groups at the end of the SAM layer were activated, and the activated molecular layer was washed with plenty of ultrapure water and dried in nitrogen stream. Then, it was immediately incubated in 40 μg/mL HSA solution at 4 °C for 12 h to ensure that the protein could be solidified on the SAM. After incubation, the surface was lightly rinsed with ultrapure water and gently blown dry with nitrogen, followed by a reaction of the cured protein layer in PBS or TKI solution for 12 h. All TKIs were prepared as a 1 mM stock solution in PBS containing 5% dimethyl sulfoxide, stored at −20 °C, and diluted before each experiment. The prepared substrates were immediately subjected to assay experiments to avoid contamination of the protein layer.

### 2.4. Functionalization of Gold-Coated Tips

The AFM probes were functionalized in the same way as the substrates; that is, first a thiol-containing SAM layer was produced, then the carboxyl terminus of the SAM layer was activated by EDC-NHS chemistry, and finally, the protein was immobilized on the tips.

### 2.5. AFM Imaging and Force Measurements

All imaging and force measurements were performed using a JPK Nanowizard II atomic force microscope (Berlin, Germany). Adhesion events and force spectroscopy experiments were performed using a functionalized probe (ContGB-GT) with a standard spring constant of 0.2 N/m and a resonant frequency of 13 kHz (the actual spring constant was determined by the thermal method) [29]. The tip radius of the probe was 25 nm and the thickness of the gold layer was 70 nm. The morphology of HSA in different samples (imatinib, nilotinib, dasatinib, bosutinib, and ponatinib) and PBS was characterized by a bare probe of the same specification in non-contact mode at a scanning rate of 1 Hz and a loading rate of 1.0 × 10^5^ pN/s. The scanning area was 1 μm × 1 μm, and the resolution was 512 × 512 pixels. The interaction forces of individual HSA pairs in different TKIs and PBS buffers were measured using functionalized probes in contact mode. For five groups of samples to be tested, six points were randomly selected in each group, and each point was repeated 50 times to measure the adhesion force events. That is, 300 force–distance curves were obtained for each group of samples to be tested.

Dynamic force spectroscopy (DFS) experiments were carried out at room temperature using the same functionalized probe and sample concentrations as the adhesion events. For different samples to be tested, the approach speed of the probe was set to 500 nm/s, at which the tip of the needle approached different positions of the sample. The dwell time was 0.5 s to ensure the occurrence of binding events [30], and the retraction speeds were set to 100, 300, 500, 700, 900, 1100 and 1300 nm/s. The force–distance curves were then determined 300 times for each set of samples in the same manner at each retraction speed in order to analyze the kinetic parameters of the interacting molecules.

### 2.6. Molecular Docking

Based on a Lamarckian genetic algorithm implemented in AutoDock4.2 [31], a computational molecular analysis of TKIs in interaction with HSA was performed. From the Protein Data Bank, we obtained the crystal structure of HSA (PDB ID: 1BM0). TKI structures were portrayed with ChemBioDraw Ultra 14.0 and ChemBioDraw 3D Ultra was used to convert the 2D structures into 3D models. In order to dock the molecules, the low-energy ligand conformations optimized by the MM2 molecular mechanics configuration optimization module that comes with the software were used. According to the literature, the default parameters were set to an initial population size of 150, 100 search runs were set for each binding site [32], and the maximum number of energy evaluations and the maximum number of generations were 250,000 and 27,000 [33], respectively. The results were clustered with the root-mean-square deviation of 2 Å. The docking results were visualized and analyzed using Discovery Studio 2016 and Pymol 2.5.

### 2.7. Statistics

Statistical analysis was performed using the software IBM SPSS Statistics 26. Data are shown as the mean with standard deviation (SD), from at least three independent experiments. Statistical significance was analyzed using a paired sample *t*-test and all data were tested for Gaussian distribution. Statistical differences are expressed as *** *p* < 0.001 and ** *p* < 0.005.

## 3. Results and Discussion

### 3.1. Adhesion Event Measurement

Using the SMFS mode of AFM as a biosensing technique, the interactions between HSA pairs were examined at the single-molecule level in the presence of different TKI solutions. In AFM-based force spectroscopy, the force–distance curves (Figure 1) are created primarily by advancing and retracting a rigid stylus over the sample and recording the force and distance traveled to deflect the cantilever, thereby recording the intermolecular interaction forces, which include van der Waals forces, hydrophobic interactions, and electrostatic interactions [34]. During each force measurement, the AFM records the approach and retraction force curves and can convert the cantilever deflection into force using Hook’s law (*f* = *k*Δ*z*, with Δ*z* being the cantilever deflection and *k* being the cantilever spring constant) [35]. Among them, the retraction curve (Figure 1, solid line) quantifies the adhesion force between the molecules modified on the probe and the substrate.

In force measurements, since the binding of molecules modified on the tip to molecules modified on the substrate is random, not every measurement has a specific bond, and it is possible that the interaction force is not detected, and it is also possible to measure the strength of multiple bonds at once [36], so the determination of the adhesion force usually requires several hundred data measurements to reduce errors. Under the same test conditions, force histograms were plotted with 300 repeated measurements for each group of samples and further fitted with Gaussian distributions to determine the adhesion force distribution of proteins in the presence of PBS and TKI solutions. As shown in Figure 2, the rupture force maximum of HSA in PBS occurred at 868.9 ± 18.2 pN, while in imatinib, dasatinib, nilotinib, bosutinib, and ponatinib solutions, peaks appeared at 450.4 ± 12.3, 524.0 ± 10.8, 425.8 ± 8.5, 368.6 ± 5.4, and 543.8 ± 21.9 pN, respectively. The results show that the addition of TKIs significantly reduced the adhesion force between protein molecules, and the decline rates were 48.16%, 39.68%, 51.02%, 57.56%, and 37.42%, respectively. In addition, due to the complex interactions between molecules, a single rupture event may correspond to different numbers and different types of bonds, which is the reason for the wide distribution of adhesion values. Among them, the range of adhesion force distributions in the control solution was 0.0–1800 pN, while in the presence of TKI, except for dasatinib, the adhesion distribution range was more dispersed, and the adhesion distribution range of other drugs was reduced, which may be related to the different spatial structures of small-molecule drugs [37]. Therefore, for the force histograms, the distribution of adhesion forces includes both specific and non-specific interactions.

Chen et al. proposed in a study that Poisson statistical methods can be used to analyze force studies in AFM [38]. Nehal I et al. also demonstrated in a scientific study on the interaction between Escherichia coli and silicon nitride that analysis of AFM adhesion data using Poisson statistics could separate specific and non-specific interactions [39]. In addition, by applying the Poisson distribution method to the force curve data in AFM, not only the adhesion properties of the samples to be tested can be obtained, but also the method is not affected by noise and can accurately determine the strength of a single bond. The main theoretical background behind the application of the Poisson distribution to the treatment of adhesion forces is that the total adhesion force obtained from the force curves in AFM is the sum of multiple discrete bonds with a finite number of interacting molecules, and within a fixed contact area [40]. Therefore, the specific interaction forces *F_i_* and non-specific interaction forces *F*_0_ of the HSA pairs can be decoupled using the following formula.
*σ*^2^*_m_* = *μ_m_F_i_* − *F_i_F*_0_(1)

For our measurement, we can plot the average value (*μ_m_*) as the abscissa and the variance (*σ^2^_m_*) as the ordinate according to formula (1), and then the *F_i_* (corresponding to the slope) and the *F*_0_ (corresponding to the intercept) of the HSA pairs can be obtained.

The dataset processed using the Poisson method included the adhesion forces of HSA molecular pairs in five TKIs and control solutions (PBS). In this case, 300 data points were collected for each set of samples to find the mean and variance of the adhesion forces, and then *F_i_* and *F*_0_ were further calculated using the Poisson formula. Table 1 summarizes the results of the mean value (*μ_m_*) and variance (*σ^2^_m_*) for PBS as an example. With the mean value as the horizontal coordinate and the variance as the vertical coordinate, Figure 3 gives a good linear relationship between them. From the slope and intercept in Figure 3, *F_i_* and *F*_0_ for HSA pairs in TKIs and PBS solutions can be obtained (Figure 4). Moreover, the paired samples *t*-test in SPSS Statistics 26 software was used to statistically analyze the specific interaction forces and non-specific interaction forces in drug solutions and controls. The results show *p*-values less than 0.005 for dasatinib and ponatinib, and less than 0.001 for imatinib, nilotinib, and bosutinib, indicating that the differences were statistically significant. It can be considered that the addition of drugs has a certain impact on the interaction between proteins. According to the literature [41], AFM was used to measure the pull-off event of the complex in which the ligand and receptor were immobilized on the tip and the substrate, respectively, and the breaking force between a single pair of molecules was generally less than 200 pN. In our experiments, the *F_i_* of HSA pairs in the control solution was 47.7 ± 1.5 pN. However, in the presence of imatinib, nilotinib, dasatinib, bosutinib, and ponatinib, *F_i_* between HSA pairs decreased sequentially to 35.1 ± 1.8, 34.1 ± 3.5, 37.3 ± 1.0, 29.1 ± 1.4, and 40.4 ± 1.9 pN, with corresponding decline rates of 26.4%, 39.9%, 21.8%, 39.0%, and 15.3%, respectively. This result is in the same order of magnitude and comparable in numerical value to the values reported in the study of individual receptor-ligand breaking forces by Desmeules et al. [42]. The *F*_0_ between HSA pairs in TKI solutions also showed a decreasing trend compared to the *F*_0_ of 116.1 ± 1.3 pN in the control solution, decreasing to 101.2 ± 2.1, 103.4 ± 1.5, 111.9 ± 1.1, 113.7 ± 2.6, and 107.8 ± 1.0 pN, respectively, with decreases of 12.8%, 10.9%, 3.6%, 2.1%, and 7.1%, respectively. The interaction forces between ligand and receptor may include hydrogen bonding, electrostatic interaction, hydrophobic interaction, and van der Waals interaction [43]; the HSA molecular pairs studied in this paper did not form chemical bonds in this system, so the specific interaction forces extracted using Poisson statistical method were mainly hydrogen bonding forces. Since tryptophan (Trp), tyrosine (Tyr), and phenylalanine (Phe) residues are very sensitive to changes in the local environment of serum albumin such as biomolecule binding [44], it can be inferred that when small molecule compounds interact with human serum proteins, the specific interaction forces between proteins are changed. In most cases, non-specific interactions always involve several types of forces existing simultaneously, among which the main ones are van der Waals forces and electrostatic interactions. Most TKIs are weakly basic drugs, and HSA is also basic under the measurement conditions because its isoelectric point is 4.7, which is lower than the experimental environment of pH 7.4, so the ionic strength and ionic species in solution remain essentially unchanged and have little effect on the non-specific interaction forces between proteins.

### 3.2. HSA Imaged by AFM in Solution

Because of its nanometer-scale spatial resolution, both lateral and vertical [45], AFM can probe the surface topography in high detail both qualitatively (via surface images) and quantitatively (via mathematical quantities such as surface roughness). Furthermore, we can enhance our understanding of the mechanics of HSA pairs by combining force measurements and morphological detection. Protein samples were operated in non-contact mode due to the risk of being damaged by the sharp AFM tip. In this way, the tip oscillates as it passes over the surface, thereby virtually preventing any contact between it and the sample, and no risk of dragging, deforming, or scratching it exists. From Figure 5, we can clearly see that the morphology of the protein has changed. The shape of HSA in the control solution was similar to small round particles, and the protein layer was uniform and stable, with a clear outline. However, in the solution of TKIs, the protein particles became larger, the boundaries became irregular, and the imaging resolution decreased. The changes in the morphology of HSA indicate that the introduction of TKIs resulted in protein aggregation or lateral diffusion. Moreover, similar regularities are also reflected in the 3D images (Figure 6). We must note that the proteins form clusters by lateral diffusion, rather than by accumulating one upon the other, since the height of each cluster is approximately as high as one HSA molecule, but the lateral dimensions are much greater [45]. Comparing the morphological changes of HSA in the six groups of solutions in Figure 5 and Figure 6, we found that the morphologies were similar. In order to further quantify the changes in the proteins in the solution, we used Image 4.7 software to process and obtain the height and roughness of the proteins. Among them, the average height (*μ*) and average roughness (*s_a_*) were obtained by the following equations:$$\mu =\frac{1}{MN}\overset{M-1}{\underset{k=0}{\sum }}\overset{N-1}{\underset{l=0}{\sum }}Z\left(X_{k},Y_{l}\right)$$
$$s_{a}=\frac{1}{MN}\overset{M-1}{\underset{k=0}{\sum }}\sum_{l=0}^{N-1}\left|z\left(X_{k},Y_{l}\right)-\mu \right|$$

Here, *M*, *N*, and *Z*($X_{k},Y_{l}$) are the length, width of the scanning area, and *Z*-axis height of the coordinate point ($X_{k},Y_{l}$).

Table 2 shows the calculated results of the morphological characterization of the HSA layer in six groups of solutions. The average height of HSA in PBS solution was 15.42 ± 0.13 nm. When small molecule drugs were added, the height of the HSA layer decreased slightly by about 1 nm. Among them, the HSA changed most significantly in the nilotinib solution, which is consistent with the 3D images. We also noticed that the roughness of the HSA layer decreased in the TKI solution, which was compared with the 2D morphology of the HSA layer in Figure 5, and we found that the larger the protein particles were, the smaller the roughness was, which might be related to the continuous lateral diffusion of the protein. In addition, we continuously observed the morphology of the HSA layer in the PBS solution and found no obvious aggregation and deposition, indicating that the changes in protein layer height and roughness were mainly caused by the introduction of small molecules. According to the literature, when TKIs bind to the polar groups of HSA, this decreases the activity coefficient of the protein and increases the hydration effect of the protein, which leads to the aggregation and lateral diffusion of the HSA layer [46].

### 3.3. Determination of Dynamic Force Spectrum (DFS) of HSA

According to widely accepted theories, specific unbinding forces are affected by intrinsic interactions among molecules and by loading rates [19]. With the help of DFS, kinetic parameters can be easily determined from AFM measurements. Detailed information about the dissociation dynamics of the interaction between proteins can be obtained by calculating the most likely rupture forces at different loading rates. The dynamic testing process occurs when the probe has experienced the dynamic process of approach-pause-retraction. In order to explore the interaction of loading rate on HSA pairs, we carried out force measurement experiments at seven different loading rates by changing the retraction rate and obtained the rupture force at different loading rates (Table 3 with control to represent). The loading rate was obtained by multiplying the retraction speed of the probe by the spring coefficient of the cantilever (0.2 N/m). According to the literature, the Bell–Evans model is suitable for expressing the relationship between the rupture force *F* and the logarithm of the loading rate *r* [47]:(2)$$F=\frac{k_{B}T}{x_{\beta }}I_{n}\left(\frac{{rx}_{\beta }}{k_{off}{\;k}_{B}T}\right)$$
where *x_β_* is the energy barrier width, *k_off_* is the dissociation rate constant, *k_B_* is the Boltzmann constant, and *T* is the absolute temperature.

According to Table 3, the rupture force between proteins increased gradually as the loading rate increased when the retraction velocities varied within a certain range. We chose a small range of loading rates because the loading rate for a typical receptor-ligand separation was also as low as 10–60 nN/s under physiological conditions [48]. The dynamic force spectrum for the interaction between proteins can be obtained by plotting the logarithm of the loading rate as the abscissa and the rupture force as the ordinate (Figure 7). Fitting the dataset according to the Belle model allowed us to obtain the kinetic parameters of the protein interactions (Table 4). The results show that the protein interactions show only a single energy barrier over the range of loading rates studied [49]. In PBS solution, the dissociation rate constant (*k_off_*) and the energy barrier width (*x_β_*) of HSA pairs were 0.106 s^−1^ and 0.063 nm, respectively. However, in the presence of TKIs, the dissociation rate constant increased and the value of *x_β_* decreased slightly. The gathered *k_off_* values for HSA complexes ranged from 0.106 s^−1^ (control conditions) to 0.256 s^−1^ (in the presence of imatinib inhibitor). These data are in agreement with previous reported biological systems such as ferredoxin NADP^+^ reductase: NADP^+^ (*k_off_* = 0.105 s^−1^) [50]; aggrecan: aggrecan (*k_off_* = 0.127 s^−1^) [51]; concavalin A: carboxipeptidase A (*k_off_* = 0.170 s^−1^) [52]; or lectin: α-GalNAc (*k_off_* = 0.680 s^−1^) [53]. This finding evidences the transient nature of the bonds involved in HSA complex under the five studied conditions. In addition, it shows that the introduction of the drug decreases the interaction force between the proteins and the stability of the protein complexes. This is consistent with the results obtained from the adhesion assay.

When a drug binds to a protein, it affects the interaction force between the proteins, thereby affecting the stability of the protein complexes. Comparing the dissociation rate constant of HSA pairs in different TKI solutions in Table 4, we found that the HSA pairs in nilotinib had the smallest dissociation rate constant of 0.197 s^−1^, indicating that the protein complexes were more stable in this solution than in several other solutions. It has been demonstrated that nilotinib has benzenoid properties as well as non-polar groups that interact with hydrophobic residues of proteins [10], so it is inferred that when it binds to proteins, it affects the hydrophobic structure of the protein itself, thus making the protein complex binding more stable. In addition, we found that the HSA pairs have a greater dissociation rate constant but a shorter lifetime in imatinib solution, presumably because of the lower stability of the complexes, as it has been noted in the literature that imatinib has a smaller binding distance r and weaker hydrophobic interactions with HAS [54]. Moreover, the HSA pairs had the smallest *x_β_* value in bosutinib, but the dissociation rate constant was higher, indicating that the drug had less effect on protein interaction. This result may be due to the specificity of certain hydrophobic residues in the protein that are not influenced by the binding of bosutinib to the protein [55]. Nevertheless, we hypothesized that there could be significant differences in the stability of HSA combinations generated in TKI solutions. Multiple amino acids in HSA, such as arginine, glutamine, tyrosine, and intermolecular hydrogen bonds certainly affect the stability of protein complexes in TKIs. Considering that the structure of such drugs has multiple non-polar groups, it is easy to assume that such drugs react with multiple amino acids of proteins, thereby affecting the interaction force between proteins.

### 3.4. Molecular Docking Analysis

Molecular docking is a powerful computer-aided technique that can provide models of the interaction between small molecules and proteins at the molecular level, so as to better explore their relationships. In our study, AutoDock software was used for molecular docking to further study the interaction between TKIs and HSA in order to gain deeper insights into their binding mode. According to the literature, in addition to binding Sudlow’s site I and Sudlow’s site II, HSA also has several other secondary sites with low or very low affinity, but most drug molecules preferentially bind to site I and site II of HAS [56]. Therefore, molecular docking is mainly studied at these two sites. Through multiple docking of the binding sites, it was found that TKI drugs were more inclined to bind to binding site I (subdomain IIA) of HSA. Site I of HSA has three hydrophobic sub-cavities, two polar plaques, and a continuous non-polar cavity [57]. This inclusive pocket can be combined in different ways with compounds of different sizes and structures. The docking results show that the lowest binding energies of the five TKIs at this site (site I) were imatinib (−7.66 kcal/mol), bosutinib (−5.73 kcal/mol), dasatinib (−5.84 kcal/mol), nilotinib (−6.92 kcal/mol), and ponatinib (−7.85 kcal/mol). Figure 8 shows the docked conformations of different TKIs with HSA. The results show that the compounds bind to proteins mainly through hydrogen bond interactions, and the amino acids Asp, Arg, Tyr, and Gln play a major role in the binding process. In addition, other non-local interactions, such as π-hydrophobic interactions and non-classical hydrogen bonding (carbon-hydrogen bond), also play a role in the binding.

Among these five TKI drugs, imatinib and nilotinib have highly similar molecular structures. Thus, as can be seen in Figure 8, in addition to forming hydrogen bonds with amino acids, where Asp108 and Arg145 form hydrocarbon bonds with imatinib, and Leu112, Ser193, and Pro113 form hydrocarbon bonds with nilotinib, the benzamide, phenyl, and imidazole portions appear to be surrounded by hydrophobic cavities of the protein [10], binding to the protein and making the bound conformation more stable. In addition, ponatinib is surrounded by the following amino acid residues: Asp 108, Pro 147, Ser 193, Gln 459, Asn 429, Val 456, Val 455, Lys 436, and Asn 109. These amino acids create a hydrophobic environment for ponatinib and increase the hydrophobic interactions between the compound and the protein. At the same time, Try452 forms a traditional hydrogen bond with the fluorine atom in ponatinib. We also found a similar hydrophobic environment around dasatinib and bosutinib and a small amount of Pi-cation interaction in the docking conformation of both drugs with proteins, mainly related to Arg145, Lys190, and Glu425. These results are consistent with the force results measured by AFM and further illustrate the dominance of hydrogen bonding and hydrophobic interactions in the effect of drugs on proteins.

## 4. Conclusions

In this study, we successfully characterized and measured the interactions between HSA pairs in different TKI solutions using AFM. The results of this study give us further knowledge and insights into the effects of exogenous compounds on PPIs. The specific and non-specific interaction forces between HSA pairs in different solutions were quantified based on the measurement of adhesion events. The results show a significant decrease in specific interaction forces and a slight decrease in non-specific interaction forces between HSA pairs in TKIs compared with the control group. The imaging of HSA under five drugs was examined, and it was shown that the effect of TKIs on HSA manifested as the aggregation of protein particles. In addition, the comparison of the dissociation rate constant with the control group shows that the dissociation rate constant of the HSA pairs in the TKI solutions increased and the stability of the protein complexes decreased due to the binding of TKIs to HSA. These phenomena are mainly attributed to the interaction of non-polar groups in the structure of such drugs with hydrophobic residues in proteins. Molecular docking assistance verified these results, showing that among the various forces, hydrogen bonding and hydrophobic interactions are the main forces for drugs to affect protein–protein interactions. In conclusion, the use of atomic force spectroscopy to study biologically relevant interactions between proteins at the single-molecule level will provide many important parameters that will help to probe the mechanism of action of PPIs in the presence of exogenous compounds or different liquid environments, and will also provide a certain reference for new drug development and disease treatment.

## Figures and Tables

**Figure 1 biomolecules-12-00819-f001:**
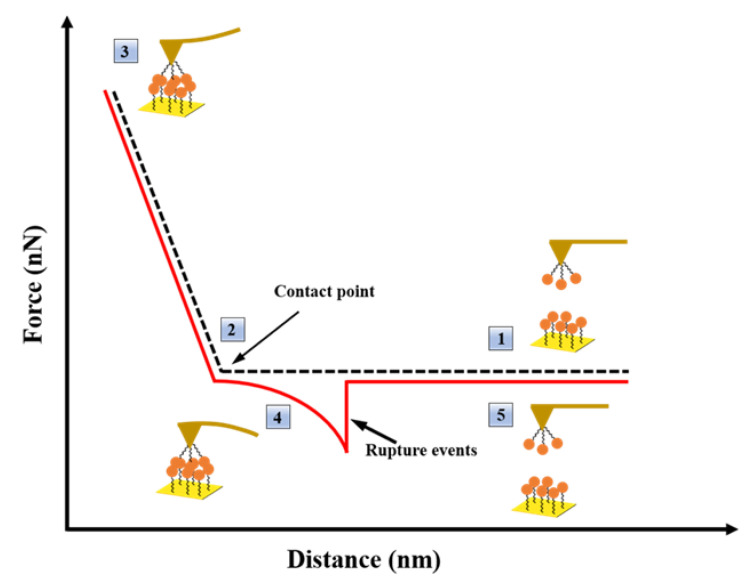
AFM force–distance curve example. The approaching stage is depicted by a dashed line, whereas the withdrawing stage is depicted by a solid line.

**Figure 2 biomolecules-12-00819-f002:**
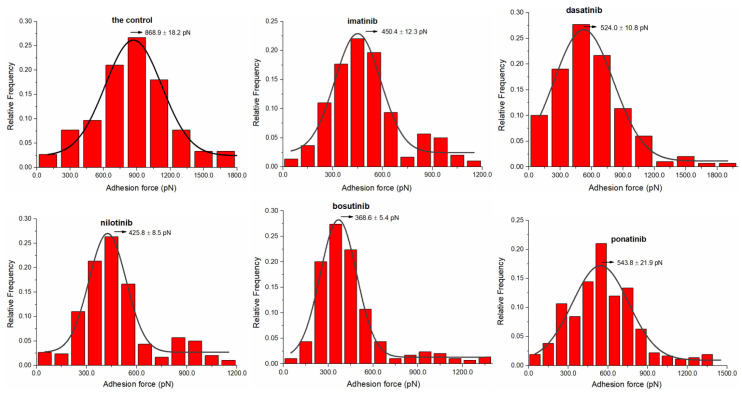
Adhesion force–frequency distribution histograms with Gaussian fitting curves recorded between HSA pairs in the control and TKIs.

**Figure 3 biomolecules-12-00819-f003:**
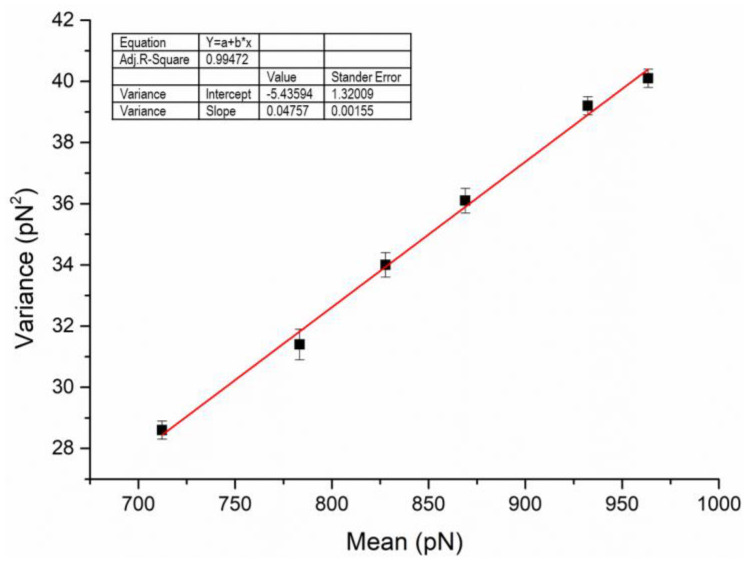
Plot of the mean (*μ_m_*) vs. variance (*σ*^2^*_m_*) for the data in Table 1 with linear regression. Each point represents a dataset collected at one of six randomly chosen sites. (R^2^ = 0.99472).

**Figure 4 biomolecules-12-00819-f004:**
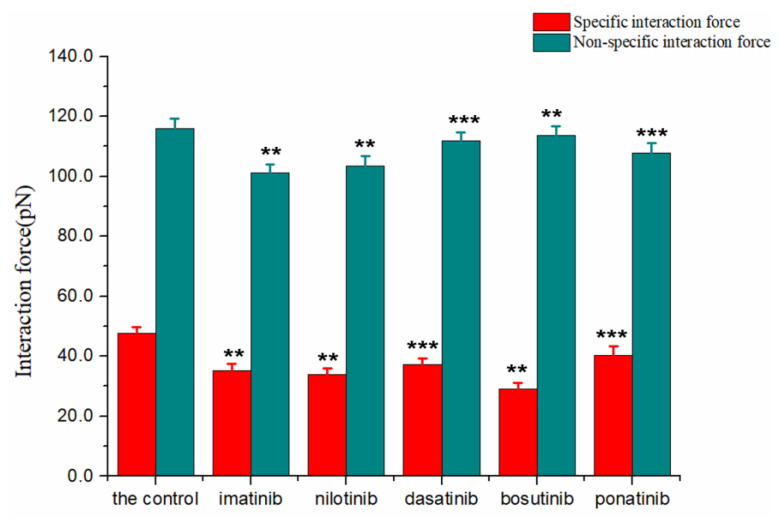
The specific/non-specific interaction forces between HSA pairs in the control and TKIs are summarized using a bar graph. All experiments were performed 300 times (*n* = 300). *** *p* < 0.001, ** *p* < 0.005 vs. the control.

**Figure 5 biomolecules-12-00819-f005:**
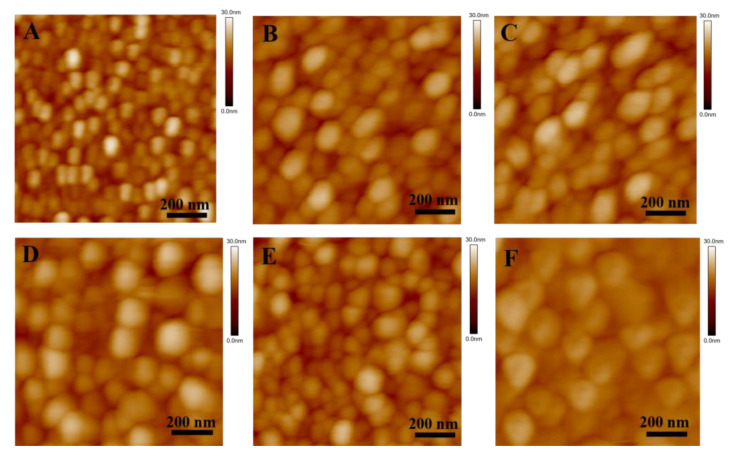
Morphology of HSA in different solutions characterized by AFM. (**A**) The control, (**B**) imatinib, (**C**) dasatinib, (**D**) nilotinib, (**E**) bosutinib, (**F**) ponatinib. Image size is 1 × 1 μm^2^; the scale bar is 200 nm; height scale is 30 nm from dark to bright.

**Figure 6 biomolecules-12-00819-f006:**
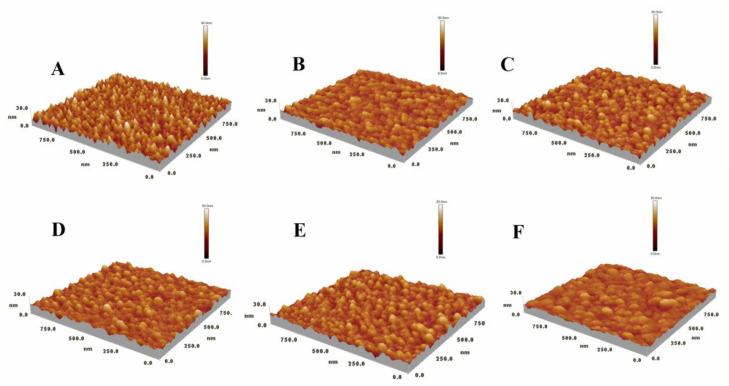
AFM measurements of the 3D-topography of HSA in various solutions. (**A**) The control, (**B**) imatinib, (**C**) dasatinib, (**D**) nilotinib, (**E**) bosutinib, (**F**) ponatinib. All images are of the same size and the high bar represents 30 nm.

**Figure 7 biomolecules-12-00819-f007:**
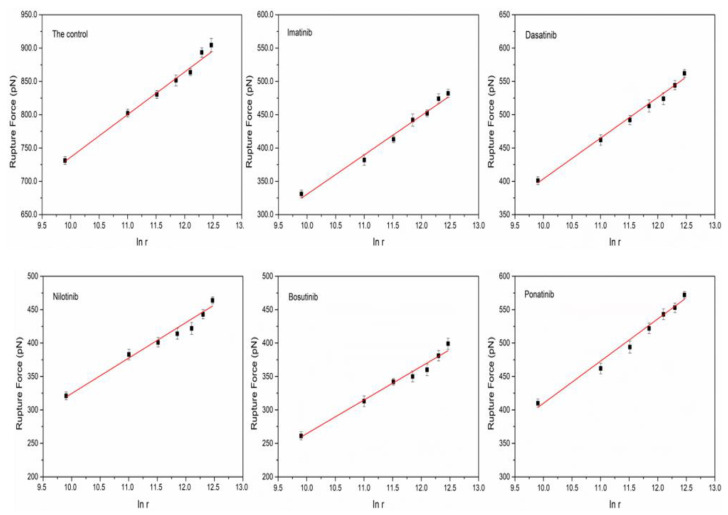
Plots of the most probable rupture forces against the logarithm of loading rates *I_n_*(*r*).

**Figure 8 biomolecules-12-00819-f008:**
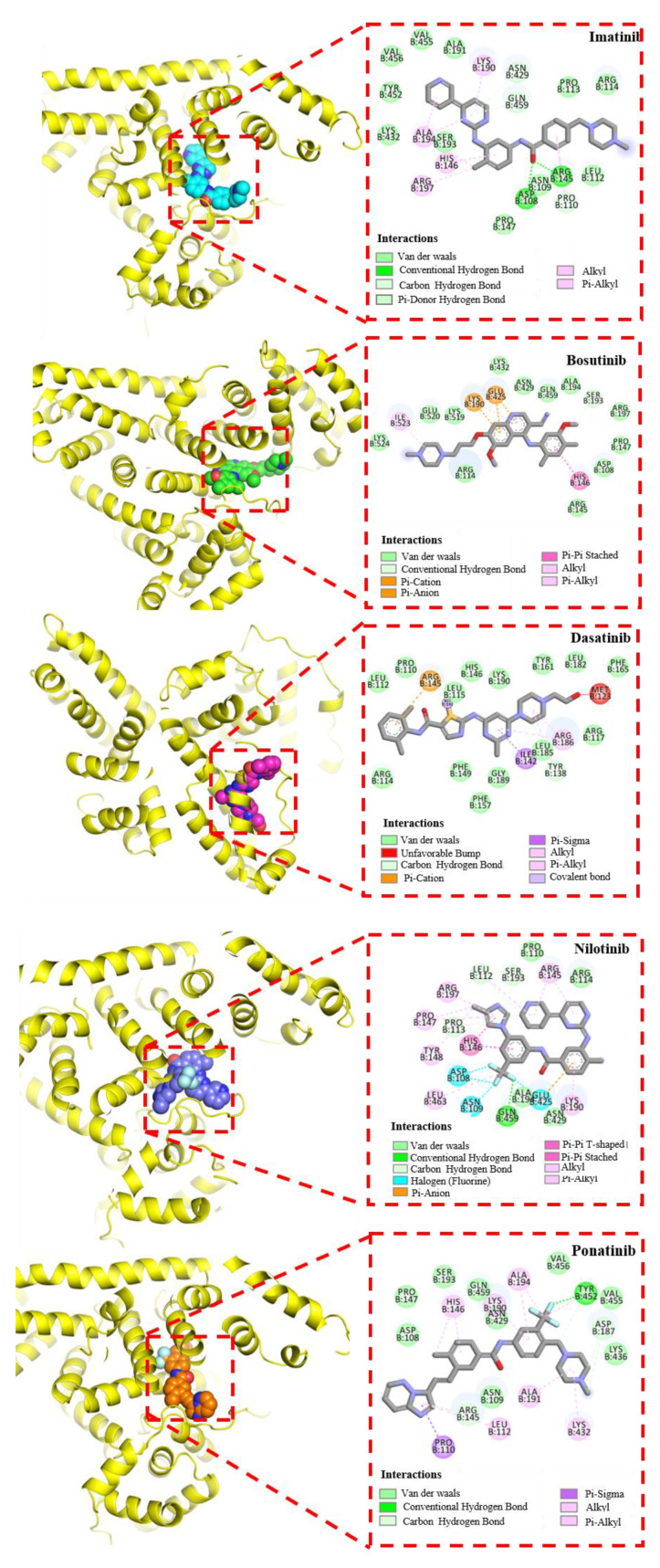
Predicted binding orientation of TKIs (rendered in spheres with lowest docking energy) in site I of HSA. The hydrogen bonding (green dotted lines) between the amino acid residues and TKIs is highlighted in the binding site.

**Table 1 biomolecules-12-00819-t001:** Results of the adhesion forces between HSA pairs in the control (PBS).

Location	*μ_m_* (pN)	*σ^2^_m_* (pN^2^)	Size of Set (N)
1	712.2	28.6	47
2	783.3	31.4	49
3	827.7	34.0	50
4	868.9	36.1	45
5	932.2	39.2	48
6	963.4	40.1	47

**Table 2 biomolecules-12-00819-t002:** Results for the average height (*μ*) and average roughness (*s_a_*) of the HSA layers in different solutions.

Sample	*μ* (nm)	*s_a_* (nm)
The control	15.42 ± 0.13	1.640 ± 0.023
Imatinib	14.30 ± 0.07	1.311 ± 0.020
Dasatinib	14.20 ± 0.12	1.318 ± 0.013
Nilotinib	14.07 ± 0.15	1.302 ± 0.018
Bosutinib	14.11 ± 0.09	1.330 ± 0.021
Ponatinib	14.19 ± 0.16	1.298 ± 0.015

**Table 3 biomolecules-12-00819-t003:** The probe retraction velocity, loading rate, and rupture force under different loading rates in the control.

Group	Retraction Velocity	Loading Rate	Rupture Force
(PBS)	(nm/s)	(pN/s)	(pN)
1	100.0	2.0 × 10^4^	731.1
2	300.0	6.0 × 10^4^	802.3
3	500.0	1.0 × 10^5^	830.3
4	700.0	1.4 × 10^5^	851.5
5	900.0	1.8 × 10^5^	863.7
6	1100.0	2.2 × 10^5^	893.4
7	1300.0	2.6 × 10^5^	904.5

**Table 4 biomolecules-12-00819-t004:** The Bell model parameters between HSA pairs in different solutions were obtained from the *F*-*I_n_*(*r*) interaction curve.

Sample	*k_off_* (s^−1^)	*x_β_* (nm)	*τ* (s)
The control	0.106	0.063	9.433
Imatinib	0.256	0.059	3.906
Nilotinib	0.197	0.052	5.076
Dasatinib	0.200	0.060	5.000
Bosutinib	0.242	0.050	4.132
Ponatinib	0.213	0.062	4.695

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
