# Peer review of "Investigating the Effect of Tyrosine Kinase Inhibitors on the Interaction between Human Serum Albumin by Atomic Force Microscopy"

_biomolecules, 2022, doi:10.3390/biom12060819_

Round 1

Reviewer 1 Report

Abstract: I would suggest the Authors to slightly rearrange the abstract. Also terms as “the specific force F1” may not be familiar to everyone.

L34: The authors citied the Ershov article reffering to the effect of the Isatin on the stability of protein complex. However, the alteration of this drug on amyloid bind in the cell was reported there from another source and not discussed. Moreover, in my opinion the Authors should refer to newer publications related more with TKIs i.e: https://doi.org/10.1080/07391102.2020.1772880; http://dx.doi.org/10.1080/07391102.2015.1089187 or https://doi.org/10.3390/molecules27041265.

L55: Maybe the Author can refer to more common methods to analyze protein–protein interactions e.g. immunoprecipitation, pull-down assay or tandem affinity purification-mass spectrometry (MS) in that section?

L59-63, L70-73: These sentences are a little too long in my opinion, and thus difficult to perceive.

L87: What was the concentration of the PBS buffer? 10mM?

L106: In my opinion, the final concentration of HSA is too high to characterize the morphology by AFM.

Figure 5&6. The quality of the AFM images is not the worst, but there is a lack of scale bars and the description is very laconic.

Table 2. I’m not convinced that the Authors were able to distinct differences of 1 nm having a tip error of 3 nm.

L378: Molecular docking analysis is too general. More details i.e. binding energy should be provided.

Reviewer 2 Report

The submitted manuscript titled: “Investigating the effect of tyrosine kinase inhibitors on the interaction between serum human albumin by atomic force microscopy” by Fu, Y.; Wang, J.; Wang, Y.; and Sung H. is an interesting work using the atomic force microspy (AFM) technique, specially dynamic force spectroscopy (DFS) operational mode to address the intermolecular interactions between human serum albumin (HSA) proteins in presence of five different tyrosine kinase inhibitors at the single molecule level. The most relevant findings of this research can serve to design more efficient drug therapies against diseases where HSA is involved as carcinomas or ischemic heart diseases, among others. The gathered findings may be relevant for the examined field. The results achieved are well-discussed during the main body of the reported manuscript. The scientific paper is well written. In my opinion the present manuscript is innovative and the methodological approached used matches with the scope of Biomolecules. For the above described reasons, I recommend the publication in Biomolecules once the following remarks will be fixed:

--------

ABSTRACT

I may recommend to erase the abbreviation “protein-protein interactions (PPis) (line 12) because it exists the potential risk to misunderstand this term with “protein phosphatase inhibitor” which also contains the same abbreviation. Authors could use “protein-protein interactions”, “intermolecular protein forces/events”, “protein adhesion forces” to prevent the repetition.  

--------

INTRODUCTION

The introduction is clearly explained and well-structured. Authors should include some previous examples were AFM-force spectroscopy were successfully devoted for biological systems.

The following statement should be added after line 63: “DFS has successfully employed to address the intermolecular forces which govern biological systems as carbohydrates:proteins [1], lipid membranes:proteins [2], proteins [3], flavoenzymes [4] DNA strands [5], RNA strands [6] and lignocellulosic biopolymers [7], among others”.

[1] Reiter, Scherer, V.; et al. Force Spectroscopy Shows Dynamic binding of Influenza Hemagglutinin and Neuraminidase to Sialic Acid. Biophys. J. 2019, 116, 1577. https://doi.org/10.1016/j.bpj.2019.03.032.

[2] Zocher, M.; et al. Single-molecule force spectroscopy from nanodiscs: an assay to quantify folding, stability, and interactions of native membrane proteins. ACS Nano 2012, 6, 961-971. https://doi.org/10.1021/nn204624p.

[3] Marcuello, C.; et al. Recognition of Proteins through Quantitative Force Maps at Single Molecule Level. Biomolecules 2022, 12, 594. https://doi.org/10.3390/biom12040594.

[4] Marcuello, C.; et al. Mechanostability of the Single-Electron-Transfer Complexes of Anabaena Ferredoxin-NADP (+) Reductase. Chemphyschem. 2015, 16, 3161-3169. https://doi.org/10.1002/cphc.201500534.

[5] Burmistrova, A.; et al. Force measurements reveal how small binders perturb the dissociation mechanisms of DNA complex sequences. Nanoscale 2016, 8, 11718-11726. https://doi.org/10.1039/c6nr02201d.

[6] Green, N.H. et al. Single-molecule investigations of RNA dissociation. Biophys. J. 2004, 86, 3811-3821. https://doi.org/10.1529/biophysj.103.026070.

[7] Marcuello, C.; et al. Atomic force microscopy reveals how relative humidity impacts the Young’s modulus of lignocellulosic polymers and their adhesion with cellulose nanocrystals at the nanoscale. Int. J. Biol. Macromol. 2020, 147, 1064-1075. https://doi.org/j.ijbiomac.2019.10.074.

--------

MATERIALS AND METHODS

Some details lack regarding the experimental AFM conditions (Section 2.5 “AFM imaging and force measurements” (line 119)). Authors should indicate the loading rate values (r) associated to the AFM tip forward velocities. Moreover, the model of the AFM probe used for AFM imaging must be also added with some specifications as the nominal tip radius.

--------

RESULTS

Result section is well-structured and clearly explained. Nevertheless, authors should pay attention to following aspects:

I)                     The authors need to take care about the significant figures of the data provided (with special attention to force data). In this framework, authors have settled four significant figures for the force data related to Figure 2 (lines 178-180), while in other manuscript sections, the significant figures related to force data decreased to one (lines 228-229 and 233-235) or two significant figures as the force data appeared on Table 3 (line 333).

II)                  The units of intermolecular adhesion forces are not homogenized. Please, the authors should pay carefully attention about this issue. For example, some force data values are shown as nN (lines 178-180), Figure 2 (line 192), Figure 4 (line 256) and Table 3 (line 333), whereas in other manuscript sections the force is depicted in pN (lines 228-229) and lines (233-234). Since the DFS measurements are carried out in liquid media I strongly encourage the authors to employ pN force units through the entire manuscript body text.

III)               Figure 2 (line 192) not only requires to be its Y-axis be modified from nN to pN units but also, the Gaussian fitting is susceptible to be improved to make easier their visualization by potential readers. Please, the authors should change the colour of the Gaussian fitting (e.g. from red to black) and also to increase the line thickness.  

IV)               “(Phe))” (line 240) and “15.42 ± 0.13 nN” (lines 296-297) contain minor mistakes to be fixed. In the first case, the second bracket should be erased. In the second point, the units are not correct. Please modify “nN” by “nm”.

V)                  The standard deviation (SD) bars should be added to Figure 3 (line 251) for each force data measurement.

VI)               Authors discuss in Figure 4 (line 256) about the analogies and differences found for the five different conditions assayed. This discussion is necessary to better understand the impact of each condition on the intermolecular interactions of HSA proteins but it lacks some robust statistical analysis. What is the statistical significance between the specific interactions of each condition (red bars)? Please, authors should incorporate further statistical analysis like Student’s test or analysis of variance (ANOVA). Please, note that this information should be also added in the respective “Materials and methods” section to explain how this calculation was made.

VII)             Figure 5 is not correct because some relevant information is hidden. The authors must add the lateral scale bar for each AFM image (as inset) and also the Z-height bar (on the right side of each AFM image). Then, I am not sure but it seems that AFM images of Imatinib and Bosutinib conditions could undergo some tip artifact (double tip effect). What do the authors comment about this point?

VIII)         The authors should also homogenize the significant figures of the AFM tip forward velocity. Table 3 (line 333) and data located in the materials and methods section (lines 136) differ on this regard.

IX)               Figure 7 (line 353) should be also modified by changing the Y-axis force data from nN to pN units as above described.

X)                  Maybe is a problem of this proof-reading version but the Figure 8 (line 412) is blurry. It could be possible to enlarge the image in order to better visualize the chemical structures of the five tyrosine kinase inhibitors.

--------

CONCLUSIONS

Authors obtained koff values in range from 0.10 s-1 to 0.25 s-1. These koff values are related to the transient phenomena of the bond rupture under load force. Some comparison should be established with other biomolecular systems with low dissociation constants (below 1.0 s-1).

The following statement should be introduced at the end of Results section (after line 350) on in Discussion section (from line 416):

“The gathered koff values for HSA complexes ranged from 0.106 s-1 (control conditions) to 0.256 s-1 (in presence of Imatinib inhibitor). These data are in agreement with previous reported biological systems as ferredoxin NADP+ reductase:NADP+ (koff = 0.105 s-1) [8], aggrecan:aggrecan (koff = 0.127 s-1) [9], concavalin A:carboxipeptidase A (koff = 0.170 s-1) [10] or lectin:α-GalNAc (koff = 0.680 s-1) [11]. This finding evidences the transient nature of the bonds involved in HSA complex under the five studied conditions”.

[8] Pérez-Domínguez, S.; et al. Nanomechanical Study of Enzyme: Coenzyme Complexes: Bipartite Sites in Plastidic Ferredoxin-NADP+ Reductase for the Interaction with NADP. Antioxidants 2022, 11, 537. https://doi.org/10.3390/antiox11030537.

[9] Harder, A.; et al. Single-molecule force spectroscopy of cartilage aggrecan self-adhesion. Biophys. J. 2010, 99, 3498-3504. https://doi.org/10.1016/j.bpj.2010.09.002.

 [10] Lebed, M.; et al. Atomic Force Microscopy and Quartz Crystal Microbalance Study of the Lectin-Carbohydrate Interaction Kinetics. Acta Phys. Pol A 2007, 111, 273-286. https://doi.org/10.12693/APhysPolA.111.273.

[11] Sletmoen, M.; et al. Single-molecule pair studies of the interactions of the alpha-GalNAc (Tn-antigen) from of porcine submaxillary mucin with soybean agglutinin. Biopolymers 2009, 91, 719-728. https://doi.org/10.1002/bip.21213.

--------

BIBLIOGRAPHY

The bibliography is not in the proper format of Biomolecules journal. Authors must take care of this aspect and deeply revise this section. For example, the reference number 1 should appear in the following form:

Bier, D.; Thiel, P.; Briels, J.; Ottmann, C. Stabilization of Protein-Protein Interaction in chemical biology and drug discovery. Prog. Biophys. Mol. Biol. 2015, 119, 10-19. https://doi.org/10.1016/j.pbiomolbio.2015.05.002.

Moreover, the amount of bibliography of the present manuscript is lower than other works with similar length in terms of pages. For this reason and in order to improve this number some advices are indicated in the upper part is this reviewing document.

--------

OVERVIEW AND FINAL COMMENTS

The submitted work is well-designed and the gathered results are interesting for the biomedical field. For this reason, I will recommend the present scientific manuscript for further publication in Biomolecules journal once all the aforementioned suggestions will be fixed.

Round 2

Reviewer 1 Report

The authors have addressed all the comments and suggestions I made in the first review. The quality of the article has significantly improved. I believe the paper is acceptable for publication in the Biomolecules. Good job!

Reviewer 2 Report

The changes made by the authors have fully satisfied my previous request. Based on the quality and importance of the submitted work I warmly recommend it to further publication in Biomolecules journal. The present article fulfills with the high-standards of Biomolecules journal.